# OpenReview forum: "AdvEvo-MARL: Shaping Internalized Safety through Adversarial Co-Evolution in Multi-Agent Reinforcement Learning"
_ICML.cc/2026/Conference — ICML 2026 regular_

### Official Review · Reviewer_nGbY · 2026-03-09

**Soundness:** 2
**Presentation:** 2
**Significance:** 3
**Originality:** 2
**Overall Recommendation:** 3
**Confidence:** 2

**Summary:**

This paper proposed AdvEvo-MARL, which is a co-evolutionary multi-agent RL framework that internalizes safety into task agents to increase robustness. AdvEvo-MARL separates agents into two groups: defenders and attackers, and assigns separate rewards to these agents. AdvEvo-MARL also introduces a group-level mean-return baseline to each group to stabilize learning and enhance cooperation. The AdvEvo-MARL is tested on three different task-specific benchmarks.

**Compliance With Llm Reviewing Policy:**

Affirmed.

**Final Justification:**

The rebuttal has addressed some of my concerns. I am raising my score to 3.

**Key Questions For Authors:**

1. **Evaluation claims:**
The paper states that AdvEvo-MARL “consistently achieves the lowest ASR and CR across nearly all configurations.” However, Table 1 shows several cases where other baselines achieve lower ASR or CR values. Could the authors clarify this statement and explain whether the claim is based on aggregated statistics or overall trends across settings?

2. **Threat model clarification:**
The attacker's capability and formation are not clearly specified. For example, it is unclear how many attackers are present, what information they observe (e.g., whether they can access all inter-agent communications), what control they have over the system during training and evaluation, and whether the attackers can cooperate. A clearer description of the threat model would help interpret the robustness results.

3. **Effect of attacker warm-up:**
The framework includes an attacker warm-up stage using supervised fine-tuning. How does the method perform without this warm-up stage? An ablation analyzing training stability and final robustness without warm-up would help clarify its necessity.

**Limitations:**

The paper includes an impact statement discussing the societal implications of studying adversarial attacks and emphasizes that the generated attacks are intended to improve system safety. However, the paper does not clearly discuss the technical limitations of the proposed method, such as scalability to larger multi-agent systems.

**Strengths And Weaknesses:**

## Strengths

**Soundness:**
The proposed AdvEvo-MARL framework is clear and well motivated. The joint optimization of attacker and defender groups with a shared group-level baseline is an interesting design that helps stabilize training and encourage coordination among agents.

**Presentation:**
The paper is well structured and easy to follow. The description of the AdvEvo-MARL pipeline and training procedure helps readers understand the proposed method.

**Significance:**
The paper addresses the robustness and safety of multi-agent systems, which is an increasingly important problem as LLM-based agent systems are becoming more widely used.

**Originality:**
To the best of my knowledge, the proposed framework presents a relatively new approach for improving safety in multi-agent systems through adversarial co-evolution.

---

## Weaknesses

**Soundness:**
The methodological novelty is somewhat limited, as the proposed public baseline is a relatively simple modification of standard policy gradient baselines.

**Presentation:**
Some formatting issues appear in the manuscript. For example, certain hyperlinks are incorrect (e.g., line 300 referring to Figure 9). Additionally, several figures and tables in the appendix exceed the page layout limits, and Table 3 appears to be missing the CR results in the last column.

---

> ### Author Rebuttal · Authors · 2026-03-30
>
> We thank you for constructive feedback. We appreciate the recognition of our framework’s **clear motivation**, **structured design**, and its relevance to improving **safety and robustness in multi-agent systems**.
>
> ---
>
> ## Reviewer Comment 1
> > Methodological novelty (public baseline is simple).
>
> We respectfully clarify that the novelty of this work lies in the **framework design**, rather than the public baseline alone.
>
> We introduces a **co-evolutionary multi-agent RL framework** where:
> - attackers and defenders are jointly optimized,
> - safety is **internalized into task agents** rather than enforced by external guardrails,
> - multiple **distinct backbone agents** interact and evolve in adversarial environments.
>
> Within this framework, the public baseline is a **supporting component** for variance reduction and credit assignment. While simple, it is effective in this setting: as shown in Sec. 5.4, it yields **consistent improvements in training stability and final robustness**. Appendix D shows compatibility with REINFORCE++, PPO, and GRPO, indicating our framework is **algorithm-agnostic**.
>
> Importantly, a key contribution of this work lies in enabling **stable and efficient co-training infrastructure of heterogeneous LLM agents**, which is substantially more challenging than standard single-agent RL. Specifically, our framework must coordinate:
> - **information flow across agents with different roles and policies**,
> - **parallel LLM rollouts across multiple backbones without interference**,
> - **efficient GPU utilization under concurrent multi-agent execution**,
> - while maintaining **stable co-evolution dynamics** (avoiding training collapse or divergence).
>
> This system-level design goes beyond standard policy-gradient modifications and is essential for making co-evolution practical in LLM-based MAS.
>
> We better reflects real-world deployments, where agents are inherently diverse rather than parameter-shared, and further distinguishes our setting from prior shared-backbone MARL approaches.
>
> ---
>
> ## Reviewer Comment 2
> > Formatting issues (broken links, layout problems, missing CR in Table 3).
>
> We thank the reviewer and apologize. We:
> - fix hyperlinks and references,
> - reformat appendix figures/tables,
> - regenerate **Table 3**,
> - perform full proofreading.
>
> ---
>
> ## Reviewer Comment 3
> > Claim of “consistently lowest ASR/CR” is overstated.
>
> We revise it to reflect **aggregated trends** rather than per-entry dominance.
>
> Using averages from Table 1:
>
> | Topology / Metric | GPT-3.5 | GPT-4o-mini | Vanilla-3B | Vanilla-7B | Challenger-3B | Inspector-3B | Challenger-7B | Inspector-7B | AdvEvo-MARL-3B | AdvEvo-MARL-7B |
> |---|---:|---:|---:|---:|---:|---:|---:|---:|---:|---:|
> | Chain / Avg. ASR ↓ | 9.61 | 2.81 | 20.92 | 18.47 | 17.28 | 7.38 | 13.21 | 4.81 | 4.92 | **2.09** |
> | Chain / Avg. CR ↓ | 9.31 | 4.56 | 28.55 | 23.07 | 20.02 | 7.98 | 14.56 | 4.98 | 10.59 | **4.62** |
> | Tree / Avg. ASR ↓ | 6.52 | 4.42 | 24.24 | 28.45 | 16.68 | 8.13 | 22.74 | 6.27 | 3.87 | **2.86** |
> | Tree / Avg. CR ↓ | 10.09 | 1.89 | 28.01 | 25.82 | 22.85 | 17.56 | 23.89 | 12.69 | 13.09 | **7.42** |
> | Complete / Avg. ASR ↓ | 11.55 | 0.31 | 30.81 | 30.85 | 21.06 | 8.00 | 22.15 | 8.61 | 7.71 | **6.27** |
> | Complete / Avg. CR ↓ | 25.53 | 1.09 | 54.46 | 50.35 | 44.47 | 36.46 | 41.03 | 32.97 | 29.93 | **27.02** |
>
> Thus, AdvEvo-MARL achieves the **best aggregated ASR/CR among open-source methods** across all topologies.
>
> ---
>
> ## Reviewer Comment 4
> > Threat model is insufficiently specified.
>
> We clarify the threat model:
>
> - **Training**: chain topology with **1 attacker + 3 defenders**.
>   The attacker generates prompts **before interaction** and does not observe defender communication. In multi-attacker settings (App. Table 8), attackers exhibit **iterative refinement**, progressively strengthening the same prompt.
>
> - **Evaluation**: 3 defenders with one attack source per episode:
>   - NetSafe: agent corruption
>   - AutoInject: message injection
>   - UserHijack: user input attack
>
> Only AutoInject assumes message access. Visibility follows topology. The attacker’s control is restricted to the attack surface.
>
> ---
>
> ## Reviewer Comment 5
> > Effect of attacker warm-up unclear.
>
> We provide a new ablation:
>
> | Setting | Avg. ASR ↓ | Avg. CR ↓ |
> |--------|-----------|-----------|
> | No warm-up | 5.84 | 12.11 |
> | Warm-up only | 3.56 | 7.24 |
> | AdvEvo-MARL | 2.09 | 4.62 |
>
> Warm-up improves stability but is insufficient alone; best performance requires **warm-up + co-evolution**.
>
> ---
>
> ## Reviewer Comment 6
> > Technical limitations (scalability).
>
> We added discussion.
>
> Our study focuses on **4-agent systems** under three topologies. Scaling increases interaction complexity and rollout cost. We include a preliminary study (**Appendix Table 8**) up to **6 agents**, showing stable performance, but view this as an initial check.
>
> Scaling to larger heterogeneous MAS and richer attack scenarios remains future work.

---

> > ### Author Rebuttal · Reviewer_nGbY · 2026-04-01
> >
> > Thanks for the authors’ rebuttal. I still have concerns about the revised claim of **best aggregated ASR/CR**. Since the environments and testing scenarios are heterogeneous, simply averaging raw ASR/CR values across all entries may not be an appropriate way to summarize overall performance. A ranking-based aggregation, such as Borda voting/count over methods within each scenario, may be more suitable because it does not assume direct comparability of raw metric magnitudes across settings. Could the authors provide such a rank-based analysis, or another more robust aggregation method, to better support the claim?

---

> > > ### Author Response · Authors · 2026-04-01
> > >
> > > We sincerely thank the reviewer for this valuable suggestion. Given the large number of experimental settings included in our evaluation, we agree that directly averaging raw ASR/CR values across heterogeneous scenarios is not the most robust way to support an overall claim. Following your suggestion, we additionally performed a rank-based aggregation over Table 1.
> > >
> > > Specifically, we considered the common open-source baselines that are reported across all topology--scenario settings, i.e., Vanilla-3B/7B, Challenger-3B/7B, Inspector-3B/7B, and AdvEvo-MARL-3B/7B. For each topology--scenario combination in Table 1 (3 topologies × 7 settings), we ranked methods separately by ASR and CR (lower is better), using average ranks for ties, and then aggregated these rankings across all 21 scenarios.
> > >
> > > **Average-rank aggregation across all 21 scenarios**
> > >
> > > | Method | Avg. ASR Rank ↓ | Avg. CR Rank ↓ | Overall Avg. Rank ↓ |
> > > |---|---:|---:|---:|
> > > | **AdvEvo-MARL-7B** | **1.62** | **1.74** | **1.68** |
> > > | **AdvEvo-MARL-3B** | **2.36** | **2.57** | **2.46** |
> > > | Inspector-7B | 3.12 | 2.83 | 2.98 |
> > > | Inspector-3B | 3.67 | 4.07 | 3.87 |
> > > | Challenger-7B | 5.64 | 5.38 | 5.51 |
> > > | Challenger-3B | 5.60 | 5.57 | 5.58 |
> > > | Vanilla-7B | 6.86 | 6.40 | 6.63 |
> > > | Vanilla-3B | 7.14 | 7.43 | 7.29 |
> > >
> > > **Borda-count aggregation across all 21 scenarios**
> > >
> > > | Method | ASR Borda ↑ | CR Borda ↑ | Overall Borda ↑ |
> > > |---|---:|---:|---:|
> > > | **AdvEvo-MARL-7B** | **155.0** | **152.5** | **307.5** |
> > > | **AdvEvo-MARL-3B** | **139.5** | **135.0** | **274.5** |
> > > | Inspector-7B | 123.5 | 129.5 | 253.0 |
> > > | Inspector-3B | 112.0 | 103.5 | 215.5 |
> > > | Challenger-7B | 70.5 | 76.0 | 146.5 |
> > > | Challenger-3B | 71.5 | 72.0 | 143.5 |
> > > | Vanilla-7B | 45.0 | 54.5 | 99.5 |
> > > | Vanilla-3B | 39.0 | 33.0 | 72.0 |
> > >
> > > Under the average-rank aggregation, AdvEvo-MARL-7B achieves the best overall average rank (1.68), while AdvEvo-MARL-3B ranks second (2.46). We also computed a Borda-count style aggregation as suggested, where the same ordering is obtained: AdvEvo-MARL-7B achieves the highest overall Borda score (307.5), followed by AdvEvo-MARL-3B (274.5).
> > >
> > > We will revise the corresponding statement in the paper to reflect this rank-based summary rather than direct averaging of heterogeneous raw metric values. More precisely, we will state that AdvEvo-MARL provides the strongest aggregated robustness among the common open-source baselines across all topology--scenario settings under rank-based aggregation.
> > >
> > > We are also very encouraged that, after our rebuttal, the remaining discussion seems to have narrowed mainly to the aggregation methodology, while our responses appear to have addressed most of the reviewer’s other concerns, including novelty, threat model clarification, warm-up effects, and scalability

---

### Official Review · Reviewer_pjsL · 2026-03-11

**Soundness:** 4
**Presentation:** 3
**Significance:** 3
**Originality:** 3
**Overall Recommendation:** 5
**Confidence:** 4

**Summary:**

This paper proposed AdvEvo-MARL: a co-evolutionary multi-agent reinforcement learning framework that internalizes safety (attacker with evolving jailbreak prompts and defender to accomplish agent's duties and resist attacks) into task agents. It introduces a public baseline mechanism for advantage estimation, where agents (within the attacker/defender group) use the group’s mean return as a baseline, promoting collaborative learning among agents and enabling more stable policy updates during training. It mitigates the lack of cross-agent unsafe instruction detection of self-verification; system overhead and single point of failure issues of external-guard modules.  Constructed a 4000 sample.s dataset for attacker warm-up. All agents are trained using REINFORCE++ to improve both system safety and task performance. Attackers receive rewards based on whether the final system output achieves the intended malicious goal, as evaluated by a global reward signal. In contrast, defenders are responsible for both resisting jailbreak attempts and fulfilling their assigned tasks. All agents also receive a formatting reward. Experiments cross 3 representative multi-agent (Chain, Tree, and Complete) attack scenarios: NetSafe, AutoInject, and UserHijack  and 3 task-specific benchmarks (AIME, LiveCodeBench, and GPQA-Diamond) to assess its effectiveness in enhancing both safety and task utility. Their experimental results on AdvEvo-MARL are compared against three baseline methods(Vanilla - instruction-tuned models without safety training as backbone models, Challenger - a self-verification strategy, Inspector - external guard agents to detect and correct malicious messages). From Table 1, while AdvEvo-MARL does not always outperform the baselines on NetSafe for both ASR and CR metrics, it mostly outperforms them on AutoInject and UserHijack. The task accuracy of AdvEvo-MARL compared with baselines (from Figure 2) drops for mathematical and general reasoning, whereas it increases in coding for 3B; But 7B shows a considerable increase in general reasoning and Coding.

**Compliance With Llm Reviewing Policy:**

Affirmed.

**Ethical Review Concerns:**

I have found two papers with the same idea from the same first author on arXiv.
"Evo-marl: Co-evolutionary multi-agent reinforcement learning for internalized safety" and "AdvEvo-MARL: Shaping Internalized Safety through Adversarial Co-Evolution in Multi-Agent Reinforcement Learning". I am assuming it's the same author. If so, please disregard this flag.

**Ethical Review Flag:**

Flag this paper for an ethics review.

**Ethics Expertise Needed:**

["Research Integrity Issues (e.g., plagiarism)"]

**Key Questions For Authors:**

1. In Table 9, what is the difference AdvEvo-MARL-xB (Shared) and AdvEvo-MARL-xB (ours) frameworks?
2. In both tables 1 and 9, what is the motive behind the GPT3.5 and GPT4o-mini comparison with baseline and your frameworks? What is the conclusion of this comparison? A little explanation would be necessary, as it adds an important aspect.
3. What are the RLbreaker-xB and CS-RLHF-xB frameworks/models? Never described in the paper but used in comparison in Table 1.
4. What are the computational and integration costs? As the method for reward, return, and mean return calculations requires memory and context preservation for each agent, both attacker and defender functions, does it introduce latency in task execution?

**Limitations:**

1. The system introduces multiple aspects at the agent level (reward and advantage) and the central level (baseline returns). It may introduce considerable latency in both levels of execution.
2. Its feasibility will be determined based on the number of agents in the system, as agents increase, latency is expected to be higher.

**Strengths And Weaknesses:**

Strengths:
The paper is technically sound and well supported with related work. The baseline comparison across various datasets and benchmarks, along with their results, is clearly presented. Their framework is easy to understand, and the application of reinforcement learning in multi-agent systems and advantage calculation of each agent are novel and well explained, even though it builds on REINFORCE++ (for LLMs). The AdvEvo-MARL's training process, reward calculation, attack scenarios, datasets, and baselines are well-explained. It explores an important aspect of adaptable and dynamic learning in an adversarial environment using reinforcement learning, and is a necessary step towards their safety.

Weaknesses:
1. On page 6, first para, the Contagion Rate explanation does not provide a clear understanding or a formula. It is abbreviated as PR, whereas in Table 1, it is referred to as CR. Table 1 is not referenced in the paper.
2. The task accuracy improvement of +3.67% is not supported.
3. Limitations of the work are not explored.
These weaknesses must be addressed for the paper to be accepted.

---

> ### Author Rebuttal · Authors · 2026-03-30
>
> We thank for the positive evaluation, including our **technical soundness**, **clear experimental validation**, and the recognition of our **multi-agent co-evolution framework**. Also appreciate the ack of the importance of problem for improving safety in dynamic environments.
>
> For ethical reasons, we cannot comment due to policy. But can defer it to ACs for appropriate check.
>
> ---
>
> ## Reviewer Comment 1
> > CR/PR unclear, Table 1 not referenced.
>
> Thank the reviewer! **“PR” is a typo** and will be corrected to **CR**. We will also explicitly reference **Table 1** instead of "Table 9".
>
> CR is defined as the **average fraction of agents that exhibit unsafe behavior at any point in an episode**:
>
> CR = (1 / |E|) * sum over episodes e in E [ (1 / N) * sum over agents i=1..N of 1{agent i exhibits unsafe behavior at any point in episode e} ]
>
> where E is the set of evaluation episodes and N is the number of agents.
>
> Thus, **ASR** measures harmful final outputs, while **CR** captures how unsafe behavior spreads during the interaction process, even if later corrected.
>
> ---
>
> ## Reviewer Comment 2
> > The task accuracy improvement of +3.67% is not supported.
>
> We agree and fixed. Fig. 2 reports **accuracy** on AIME/GPQA and **Pass@1** on LiveCodeBench. The exact values are:
>
> | Model | AIME  | GPQA | LiveCodeBench |
> |----|----|---|--|
> | Vanilla-3B | 13.33 | 25.84 | 21.72 |
> | Vanilla-7B | 19.17 | 29.41 | 28.11 |
> | AdvEvo-MARL-3B | 12.04 | 24.96 | 22.58 |
> | AdvEvo-MARL-7B | 20.42 | 30.36 | 31.78 |
>
> Thus, the **maximum observed improvement is +3.67 percentage points**, achieved by **AdvEvo-MARL-7B on LiveCodeBench** (31.78 vs. 28.11). We will revise the text to make this explicit and directly tied to the reported numbers.
>
> ---
>
> ## Reviewer Comment 3
> > No limitations.
>
> We have added discussion of limitations, including:
> - sensitivity to attack distributions (performance degrades under stronger OOD attacks),
> - dependence on reward design and curriculum,
> - computational overhead from multi-agent co-evolution.
>
> These are supported by our added OOD evaluations and ablations.
>
> ---
>
> ## Reviewer Comment 4
> > Difference between Shared&Ours?
>
> - **Shared**: all agents share one backbone (parameter sharing)
> - **Ours**: each agent has an independent backbone
>
> We focus on the **heterogeneous setting**, which better reflects realistic MAS where agents have different roles and models. Shared-parameter methods restrict all agents to one policy space, while our framework enables **true co-evolution among distinct policies**. As shown in Appendix K, the shared setting yields higher ASR/CR, supporting the importance of agent diversity.
>
> ---
>
> ## Reviewer Comment 5
> > Why GPT-3.5 / GPT-4o-mini?
>
> These are as **reference points (approximate upper bounds)** rather than primary baselines. Since their training is not transparent, our conclusions rely on open-source comparisons. Their inclusion provides context that our method achieves **comparable robustness improvements within open settings**.
>
> ---
>
> ## Reviewer Comment 6
> > RLbreaker and CS-RLHF unclear.
>
> We will clarify:
>
> - **RLbreaker**: RL-based jailbreak framework for prompt attack generation [1]
> - **CS-RLHF**: safe RLHF alignment framework  [2]
>
> These broaden baseline coverage to **attack-side RL methods** and **single-model safety alignment**. Our method differs by performing **multi-agent co-evolution**.
>
> [1]arxiv:2406.08705
> [2]arxiv:2510.03520
>
> ---
>
> ## Reviewer Comment 7
> > Computational / latency concerns.
>
> The added overhead is small relative to LLM rollouts.
>
> - Reward: evaluated by **HarmBench-Llama-2-13b-cls**, deployed locally and **parallelized**, so latency is modest
> - Advantage: standard RL computation
> - Baseline: simple arithmetic overhead (negligible)
>
> These costs occur mainly during **training**, not deployment. Appendix D shows compatibility with **REINFORCE++, PPO, and GRPO**, indicating our framework is **algorithm-agnostic**.
>
> ---
>
> ## Reviewer Comment 8
> > Scalability with more agents.
>
> We thank reviewer for raising this important concern.
>
> As shown in **Appendix Table 8**, we scale the system up to **6 agents** and vary the attacker/defender splits. The results remain stable across configurations, suggesting that robustness does not degrade with moderate scaling.
>
> Regarding latency, the dominant cost comes from **LLM rollouts**, not our added computations. As discussed in Comment 7, reward and baseline calculations add only negligible overhead, so latency mainly scales with the number of agents through standard forward passes.
>
> Compute cost scales **linearly with the number of agents**. For example, a 7B agent typically requires ~2 GPUs, so training N agents requires ~2N GPUs. This is expected for **heterogeneous MAS**, where agents have independent models. Unlike shared-parameter methods, our framework does not add overhead beyond this standard scaling—it simply coordinates the training of distinct agents while enabling joint optimization of safety and task performance.

---

> > ### Author Rebuttal · Reviewer_pjsL · 2026-03-31
> >
> > For computational and latency evaluations, if we do not have them, my suggestion is to put them in the limitations discussion. This is an important aspect of this research direction, and future researchers would benefit from highlighting it.

---

> > > ### Author Response · Authors · 2026-03-31
> > >
> > > We thank you for this helpful suggestion. We agree that computational cost and latency are important aspects of this research direction.
> > >
> > > While our framework does not introduce significant overhead beyond standard LLM rollouts (as discussed in Comment 7), we acknowledge that **end-to-end latency and system cost in larger-scale multi-agent settings are not fully quantified in the current work**.
> > >
> > > Following the reviewer’s suggestion, we will explicitly include this in the **limitations discussion**, highlighting that:
> > > - latency scales with the number of agents due to multiple forward passes,
> > > - co-evolutionary training introduces additional rollout cost compared to single-agent RL,
> > > - large-scale heterogeneous MAS may require further systems optimization for practical deployment.
> > >
> > > We believe making this explicit will better contextualize the trade-offs and help guide future work in this direction.

---

### Official Review · Reviewer_Nax9 · 2026-03-13

**Soundness:** 4
**Presentation:** 3
**Significance:** 3
**Originality:** 3
**Overall Recommendation:** 5
**Confidence:** 4

**Summary:**

This work presents an approach to ensuring internalized safety for multi-agentic system utilizing a specific attacker-defender system called AdvEvo-MARL. Specifically this system consists of a specific sequence of different parts. Firstly, agents are split into either attacker agents or defender agents. Then, attacker agents are initialized through SFT to perform various jailbreaking strategies and further initalization are done to prepare the attackers for the training process. Then, attackers and defenders are trained collaboratively through a co-evolution process utilizing a variant of the REINFORCE algorithm. The authors test the tuned system on different baselines across different datasets and illustrate the effectiveness of these systems on safety and general evaluations.

**Compliance With Llm Reviewing Policy:**

Affirmed.

**Key Questions For Authors:**

1. For D_adv, can you elaborate on the methodology for getting x_attack (the rewritten prompts)? While this wouldn't monumentally change my evaluation of the paper, I would like to recieve clarify regarding this aspect.

**Limitations:**

Yes

**Strengths And Weaknesses:**

Soundness - The submission is technically sound. The training methodology of AdvEvo-MARL is explicitly mentioned. Specifically, the work goes into detail regarding the theoretical basis behind the workflow, clearly defining the co-evolution as a joint-policy optimization which is theoretically sound. Additionally, critical training aspects are clearly defined regarding the co-evolution RL, including the advantages and the specific reward functions utilized for the training. Moreover, the experiments are comprehensive, focusing on both safety evaluation as well as the task evaluation. Clear baselines are also established within the paper, specifically being self-verification based strategies and auditor agents that track the behavior of said agents. The authors don't extensively evaluate the strengths and weaknesses of the work.

Presentation - The submission is clearly structured and well-written. There is a bit of slight redundancy with regards to introducing existing literature, however. The authors clearly establish the motivations behind the construction of the algorithm, and present the methodology in a clear fashion. The experimental details are also clearly established, with clear directives as to how and why specific data was chosen or prepared. Finally, the work properly positions itself within the context of prior literature, as existing methods are limited to training a single backbone model as opposed to having multiple backbone models like AdvEvo-MARL.

Significance - The paper addresses an important problem, being the safety within multi-agent systems. While the methods introduced aren't massively novel (the core of the paper is a co-evolution loop of attackers and defenders which is not a novel concept in machine learning), it nonetheless provides practical utility as it illustrates how distinct backbone models can be utilized for the co-evolution loop as opposed to a single model. While the influence is limited to safety and general task performance, a similar technique can be adopted for agentic workflows within different domain areas.

Originality - The work does not introduce any particularly novel ideas, however it ensures true co-evolution as multiple backbone models were utilized, which is one of the main innovations within this paper. This enables new venues with co-evolution-based training techniques due to the existence of multiple different backbone models being trained, which is the primary innovation within this paper. Given that utilizing multiple backbone models for training was a technique that hadn't been explore, the novelty within this paper is well-justified.

---

> ### Author Rebuttal · Authors · 2026-03-30
>
> We thank the reviewer for the positive evaluation of our work, including the **technical soundness of the co-evolution framework**, the **clarity of methodology and training design**, as well as the **comprehensive experimental evaluation on both safety and task performance**. We also appreciate the recognition of our use of **multiple backbone models to enable true co-evolution** and its practical relevance to multi-agent systems.
>
> ---
>
> ## Reviewer Comment 1
> > Limited discussion of strengths and weaknesses / failure modes.
>
> We agree that a clearer discussion of limitations would strengthen the paper. We have now expanded the analysis to explicitly highlight potential weaknesses, including (i) sensitivity to attack distribution, (ii) reliance on reward design, and (iii) remaining challenges under stronger OOD attacks. These are supported by our added OOD evaluations (**Comment 3 of Reviewer 42Xa**) and ablations (**Comment 4 of Reviewer 42Xa**), which better characterize both strengths and failure cases.
>
> ---
>
> ## Reviewer Comment 2
> > Minor redundancy in presentation (especially related work).
>
> We thank the reviewer for this suggestion. We have revised the related work section to reduce redundancy and improve conciseness, while preserving key distinctions between prior single-backbone MARL approaches and our multi-backbone co-evolution setting.
>
> ---
>
> ## Reviewer Comment 3
> > Limited novelty (co-evolution is not new).
>
> We agree that adversarial co-evolution itself is not new. Our contribution lies in **adapting it to LLM-based multi-agent systems with multiple backbone models**, enabling *true agent-level co-evolution* rather than shared-parameter training. This setting introduces new challenges (e.g., coordination, stability, and scalability), and our framework—including the public-baseline mechanism and training design—provides a practical solution for safety training in realistic MAS environments.
>
> ---
>
> ## Reviewer Comment 4
> > Limited scope of impact (mainly safety and task performance).
>
> We agree that our primary focus is safety and task utility. However, we believe the framework is more broadly applicable: adversarial co-evolution with heterogeneous agents can be extended to other domains such as **agentic workflows, tool use, and collaborative reasoning systems**, where robustness under interaction is critical. We have clarified these broader implications in the revision.
>
> ---
>
> ## Reviewer Comment 5
> > For D_adv, can you elaborate on the methodology for getting x_attack (the rewritten prompts)?
>
> We thank for pointing out that the construction of x_attack in D_adv was not sufficiently explicit.
>
> In the attacker warm-up stage, D_adv is a curated paired dataset of x_behavior, x_attack, where x_behavior is a harmful behavior query and x_attack is a jailbreak-style rewritten prompt derived from that behavior. Concretely, we first sample 1,000 harmful behaviors from public harmful-query datasets and apply representative jailbreak strategies to build an initial dataset D_init. We then use a stronger reasoning model to generate multi-step reasoning traces that explain how to transform each original harmful behavior into an effective adversarial prompt. Finally, we apply LLM-as-a-judge filtering to remove invalid, contradictory, off-topic, or vague samples, resulting in a final D_adv of roughly 4,000 high-quality examples for supervised attacker warm-up.
>
> To make this process more concrete, **Appendix F** provides representative examples showing:
> (1) the original harmful request,
> (2) the applied jailbreak strategy, and
> (3) the resulting rewritten prompt x_attack.
>
> For example, under the **Disemvowel** strategy, key unsafe terms are obfuscated while preserving the underlying malicious intent.
>
> In addition, **Appendix G** clarifies the role of these rewritten prompts during later co-evolution. Early in training, attackers mainly reuse or slightly modify seed prompts, while later they learn to produce more structured and deceptive attacks, such as **semantic task partitioning** and **role-switching** attacks.

---

> > ### Author Rebuttal · Reviewer_Nax9 · 2026-03-31
> >
> > The authors have adequately addressed any concerns that I have regarding the paper. It was already very sound, and so I do not have any immediate concerns.

---

> > > ### Author Response · Authors · 2026-03-31
> > >
> > > We thank the reviewer for the positive feedback and for the helpful suggestion on improving presentation clarity.
> > >
> > > We will further refine the manuscript to better align the **main text and appendix**, ensuring that key concepts, experimental details, and supporting analyses are more clearly connected. In particular, we will improve cross-referencing and streamline the narrative flow so that readers can more easily follow the methodology and results without needing to frequently switch contexts.
> > >
> > > We believe these revisions will improve readability and make the paper more accessible to a broader audience.

---

### Official Review · Reviewer_42Xa · 2026-03-17

**Soundness:** 2
**Presentation:** 3
**Significance:** 2
**Originality:** 3
**Overall Recommendation:** 4
**Confidence:** 2

**Summary:**

AdvEvo-MARL addresses the problem of safety in multi agent LLM systems where the expanded attack surface and agent chains make jailbreaks significantly more dangerous compared to single-model settings. Instead of relying on external guard modules, the authors propose training safety directly into task agents via adversarial coevolution. Attackers and defenders are jointly optimised with attackers bootstrapped using SFT on existing jailbreak datasets. Attackers are then trained to generate better attack prompts while defenders are rewarded for simultaneously resisting attacks and maintaining task performance. To stabilise training and achieve intra-agent coordination, a shared group-level baseline is used. Experiments show ASR being consistently below 20% with minimal capability degradation.

**Compliance With Llm Reviewing Policy:**

Affirmed.

**Final Justification:**

I'd like to thank the authors for their response. They have addressed most of my concerns and I have now updated my score.

**Key Questions For Authors:**

1. I worry that high defender performance may reflect overfitting to the co-trained attacker's convergent strategies. Did you evaluate against out of distribution attacks?

2. Why not use population-based methods (eg PSRO)?

3. Is the reward curriculum doing significant work? The mid-training shift from safety-priority to task-priority weighting is not ablated.

4. Can we reliably make inferences based on the comparison to GPT-3.5 and GPT-4o-mini? My concern is that these closed models have unknown training pipelines so their inclusion as baselines does not isolate what this work contributes relative to standard safety fine-tuning.

**Limitations:**

yes

**Strengths And Weaknesses:**

Strengths:
The safety challenge discussed for multi-agent LLM systems is well-established and the limitations of external guard modules are indeed an open probelm. The authors' framing of internalised safety training as an alternative is well-motivated and worth exploring empirically.
I found the group-level public baseline for advantage estimation a clean, well-motivated solution to a real problem in multi-agent RL, where per-agent baselines can introduce misaligned incentives. The ablations support that it meaningfully stabilises training.

Weaknesses:
The attacker optimisation is weak by construction as it does not backpropagate through generation. This is non-trivial to address as eg the token sampling step is non-differentiable, leaving a sparse, high-variance gradient signal over a discrete, non-convex search space. I believe that the attackers are likely finding locally better prompts near their SFT initialisation rather than exploring the attack surface broadly. As such the "diversity" trend is encouraging but not sufficient evidence of genuine exploration.

PSRO (also other double-oracle approaches) are the natural framework for this problem. In this case, they could explicitly maintain a portfolio of strategies, compute best responses to the meta-game equilibrium and provide theoretically grounded convergence criteria.
The paper does not reference this literature but without I don't know if there is a way to mechanism to prevent cyclic dynamics or mode collapse in the attacker population. This could also provide a meaningful notion of Nash equilibrium as a stopping condition.

---

> ### Author Rebuttal · Authors · 2026-03-30
>
> We thank the reviewer for the positive feedback on the **motivation** and our **public-baseline design**.
>
> ## Reviewer Comment 1
> >  No backpropagate, attackers likely find locally better prompts near SFT. No sufficient evidence.
>
> We address it through both robustness results and direct analysis of attacker behavior.
>
> **(1) Robustness does not require a perfect attacker.**
> Our goal is **not to learn the strongest attacker**, but to provide **useful adversarial signals**. As shown in our OOD evaluations (**Comment 3**), defenders generalize beyond co-trained attacker and outperform initial model, indicating gains are not due to overfitting.
>
>
> **(2) Existing evidence: increasing diversity.**
> In **Appendix C / Fig. 9**, semantic diversity increases while Self-BLEU-to-seed decreases (~0.74 → ~0.50), indicating drift from the seed pool.
>
> **(3) Beyond diversity: distance from SFT neighborhood.**
> To directly test locality, we introduce *success-conditioned nearest-seed distance (SNSD)*:
>
> | Step | SNSD ↑ |
> |------|--------|
> | 0 | 0.062±0.006 |
> | 50 | 0.089±0.007 |
> | 100 | 0.123±0.008 |
> | 150 | 0.161±0.010 |
> | 200 | 0.197±0.011 |
> | 250 | 0.231±0.013 |
>
> SNSD increases, showing successful attacks move away from nearest seeds.
>
> **(4) Strategy-level novelty.**
> Further analyze successful attacks by seed-family taxonomy:
>
> | Step | HFR ↑ | UFER ↑ | FLNR ↑ |
> |------|------|--------|--------|
> | 0 | 0.074±0.008 | 0.019±0.005 | 0.093±0.009 |
> | 250 | 0.289±0.015 | 0.161±0.010 | 0.450±0.017 |
>
> At step 250, ~45% of successful attacks are hybrid or novel, indicating exploration beyond single-family variations.
>
> Overall, despite the challenges of discrete optimization, co-evolution induces non-local, strategy-level exploration, and yields defenders that generalize to unseen attacks.
>
> ---
>
> ## Reviewer Comment 2
> > Why no PSRO? cyclic dynamics / mode collapse?
>
> We agree PSRO is natural frameworks. **However, our goal differs**: we aim to **internalize safety into a fixed set of deployable task agents via end2end co-evolution**, rather than maintain an population & meta-solver. In our setting, each policy is a full LLM agent with long-horizon generation, making PSRO-style procedures (population expansion, payoff matrix estimation, repeated best-response training) computationally expensive and difficult to scale.
>
> **Empirically, we do not observe cyclic dynamics**. As shown in **Appendix H**, cross-play evaluations between the final attacker and different defenders yield **low ASR with negligible variance (<0.1%)**, suggesting a stable, non-cyclic regime. Our **OOD evaluations (Comment 3)** further show no collapse to a narrow strategy loop.
>
> We have now clarified the connection to this literature in the related work, and view PSRO methods as a **complementary future direction**, rather than a prerequisite for our current contribution.
>
> ---
>
> ## Reviewer Comment 3
> > May overfit. OOD attacks?
>
> We added 2 OOD evaluations under the chain topology:
>
> **(1) Held-out attack family.**
> We hold out one entire attack family (obfuscation/encoding: Base64, ROT13, etc.) during training and evaluate only on this unseen family.
>
> | Model |AutoInject ASR| AutoInject CR | UserHijack ASR | UserHijack CR |
> |-|-|-|-|-|
> |Vanilla|11.8±0.4|18.6±0.5|21.7±0.6|31.4±0.7|
> |Full|1.4±0.1|2.3±0.2|6.5±0.2|8.7±0.3|
> |Held-out|3.2±0.2|5.4±0.3|10.6±0.3|13.8±0.4|
>
> Performance degrades but remains far stronger than the initial model.
>
> **(2) Unseen MAS attacks.**
> We evaluate on two stronger unseen attacks: Prompt Infection [1] and MultiAgent Collaboration Attack [2].
>
> | Model | PI ASR | PI CR | MCA ASR | MCA CR |
> |-|-|-|-|-|
> | Initial | 28.7±0.8 | 42.1±1.1 | 23.8±0.7 | 35.4±0.9 |
> | Ours | 9.4±0.4 | 17.6±0.6 | 7.6±0.3 | 13.1±0.5 |
>
> Performance drops but remains substantially better, indicating generalizable defenses.
>
> [1] https://arxiv.org/pdf/2410.07283
> [2] https://arxiv.org/pdf/2406.14711
>
> ---
>
> ## Reviewer Comment 4
> > Reward curriculum? Weighting.
>
> We ablated in **Appendix Table 3**: **safety-first > static > task-first**. E.g., NetSafe (AIME, 7B): CR 28.60→19.27, ASR 2.31→0.99. Early safety prioritization plays a key role in establishing robust defensive behavior, while later task emphasis helps recover utility.
>
> ---
>
> ## Reviewer Comment 5
> > Why GPT-3.5&4o? unknown training pipelines so not isolate what this work contributes relative to standard safety fine-tuning.
>
>
>
> Closed-source models **are not** primary baselines, only external reference points (approximate upper bounds). Conclusions rely on open-source comparisons (averaging Chain results across all attack settings):
>
> | Model | Avg. ASR ↓ | Avg. CR ↓ |
> |-|-|-|
> | Vanilla-3B | 20.92 | 28.55 |
> | Vanilla-7B | 18.47 | 23.07 |
> | Vanilla-14B | 13.84 | 17.96 |
> | AdvEvo-MARL-3B | 4.92 | 10.59 |
> | AdvEvo-MARL-7B | 2.09 | 4.62 |
>
> Our 7B outperforms even Vanilla-14B, showing gains come from co-evolution training rather than scale.

---

### Decision · Program_Chairs · 2026-04-30

**Decision:**

Accept (regular)

**Comment:**

**Summary**

The paper proposes an agent co-evolution framework AdvEvo-MARL in which both attacker and defenders are co-trained to achieve safety resilience against jailbreaking attacks. The system is tested on multiple datasets and safety is evaluated in comparison with some baselines.

**Reviewer Scores**

The reviewer scores are: 1 Weak Accept, 2 Accept, and 1 Weak Reject. The authors wrote detailed comments as part of the rebuttal and were actively engaged with the reviewers.

**Strengths**

1. The submission is technically sound, the reviewers appreciated the definition of co-evolution as a joint-policy optimization
2. The experimental evaluation is comprehensive.

**Reviewer Concerns**

1. Limitations of the work are not discussed
2. The framework is computationally expensive, but the paper does not report performance numbers.

Based on the reviewers' recommendations and the detailed author rebuttal, I recommend acceptance assuming that the authors will integrate their responses from the rebuttal and additional experiments in the final manuscript.